# Planetary Health Diet and Cardiovascular Disease Risk in the Seguimiento Universidad de Navarra (SUN) Cohort

**DOI:** 10.3390/nu17010027

**Published:** 2024-12-25

**Authors:** Karen Berenice Guzmán-Castellanos, Itziar Zazpe, Susana Santiago, Maira Bes-Rastrollo, Miguel Ángel Martínez-González

**Affiliations:** 1Department of Preventive Medicine and Public Health, School of Medicine—Clínica Universidad de Navarra, University of Navarra, Irunlarrea 1, 31008 Pamplona, Spain; kguzmancast@alumni.unav.es (K.B.G.-C.); mbes@unav.es (M.B.-R.); 2Department of Nutrition and Food Sciences and Physiology, University of Navarra, Irunlarrea 1, 31008 Pamplona, Spain; izazpe@unav.es (I.Z.); ssantiago@unav.es (S.S.); 3CIBER Fisiopatología de la Obesidad y Nutrición (CIBEROBN), Instituto de Salud Carlos III (ISCIII), 28040 Madrid, Spain; 4Navarra Institute for Health Research (IdiSNA), 31008 Pamplona, Spain; 5Department of Nutrition, Harvard T.H. Chan School of Public Health, Boston, MA 02115, USA

**Keywords:** planetary health diet, cardiovascular disease, EAT-Lancet diet

## Abstract

Background/Objectives: Noncommunicable diseases, particularly cardiovascular disease (CVD), represent a significant global public health challenge, with unhealthy diets as a major risk factor. This study investigates the association between adherence to the Planetary Health Diet proposed by the EAT-Lancet Commission and CVD risk. Methods: Utilizing data from the Seguimiento Universidad de Navarra (SUN) cohort, which included 18,656 participants (mean age 38 years; 61% women), we assessed dietary intake using a validated food frequency questionnaire and the Planetary Health Diet Index to evaluate adherence (range 0–42). CVD was defined as new-onset stroke, myocardial infarction, or CVD death. Results: After a median follow-up time of 11.5 years, 220 cases of CVD were identified. Higher adherence to the Planetary Health Diet revealed no statistically significant reduction in CVD risk associated with the diet. Cox proportional hazard models indicated a trend towards lower CVD risk in the highest adherence quartile, but this did not reach significance (HR 0.77, 95% CI 0.51–1.18, p-trend = 0.127). Sensitivity analyses corroborated these results. Discrepancies in previous studies highlight the complexity of dietary assessments and underscore the need for standardized scoring systems. Conclusions: In a large Spanish cohort, adherence to the Planetary Health Diet showed no significant reduction in CVD risk. Further research is needed to reach a consensus on the operational definition of the Planetary Health Diet and to clarify the relationship between diet and CVD risk.

## 1. Introduction

Noncommunicable diseases (NCDs) represent a major global public health issue [1], and the major contributor to the climate change crisis is the modern global food system [2]. Approximately, 19.1 million deaths were attributed to cardiovascular disease (CVD) in the world [3]. In Spain, CVD is the leading cause of death from NCD [4]. Unhealthy diets are one of the main risk factors for CVD [5], which not only impacts human health but also planetary health [6]. It is well known that unhealthy diets rich in fats, sugars, red meat, and salt have negative health effects and contribute to the development of chronic diseases [7]. On the contrary, some dietary patterns, such as healthy plant-based diets (PBD), are significantly associated with a lower incidence of NCD especially CVD [8,9]. Thus, PBD has been proven excellent for the regulation of blood lipid levels, oxidative stress, inflammation, and hyperglycemia [10], becoming a major public health priority in response to world food insecurity, health, and environment [11].

In 2019, the EAT-Lancet Commission on “Healthy Diets from Sustainable Food Systems” created a sustainable reference diet that provides a set of recommendations for feeding the entire world population within planetary boundaries [1,12]. The EAT-Lancet diet or Planetary Health Diet is based on an intake of 2500 kcal per day [12] and classifies foods into two groups: those with emphasized consumption (vegetables, fruits, unsaturated oils, legumes, whole grains, nuts, and fish) and those with limited consumption (beef and lamb, pork, poultry, eggs, dairy and dairy products, potatoes, and added sugars). By adhering to these guidelines, the Planetary Health Diet could potentially prevent 11 million deaths annually among the global adult population (19% to 24% of total deaths) and offer health benefits such as a decreased risk of CVD [12,13]. The main reason may be that this diet demands an overall increase of 100% in the intake of healthy foods and a reduction of more than 50% in the intake of unhealthy foods, which would also contribute to lowering greenhouse gas emissions, as well as reducing land and water usage [14].

Recognizing its potential impact, several studies have used various indices to assess adherence to the diet proposed by the EAT-Lancet Commission and its association with CVD risk [1,2,11,15,16,17,18,19,20,21,22,23,24]. However, the evidence is scarce and limited, with discrepancies in the main findings across these studies. Some have identified a significant association between greater adherence to the Planetary Health Diet and a lower risk of CVD [2,15,16,17,18,19,20,21,22,24]. However, others have found no such association [1,11,23].

Therefore, we aimed to investigate the association between adherence to the Planetary Health Diet and the risk of CVD in the Seguimiento Universidad de Navarra (SUN) project aiming to address these discrepancies. This study is particularly noteworthy as it is the first to be conducted in the adult population of Spain.

## 2. Materials and Methods

### 2.1. Study Design and Participants

The SUN project is a dynamic, multi-purpose, and prospective Mediterranean cohort of university graduates in Spain that has been ongoing since December 1999 (http://medpreventiva.es/MvbqgK, accessed on 18 October 2024) with a retention rate of 91.22%. Participants completed a baseline self-administered questionnaire (Q0) which included 554 items to collect information about sociodemographic, lifestyle, medical history, and dietary variables. Questionnaires are sent via post or email. Additional details of the design and methodology can be found elsewhere [25].

For this study, a total of 22,899 participants recruited before September 2020 were eligible for the analysis of incident CVD, ensuring that they were able to complete at least the first follow-up questionnaire. We excluded 352 participants with pre-existing CVD, as well as 2095 participants who fell outside predefined limits of total energy intake (<800 kcal/d or >4200 kcal/d for men and <500 kcal/d or >3500 kcal/d for women) [26] and 1796 participants without follow-up. As a result, 18,656 participants were included in our final analysis (Figure 1).

### 2.2. Dietary Assessment

Dietary data from the baseline and after 10 years of follow-up were assessed using a self-administered food frequency questionnaire (FFQ), which has been validated previously, and its reproducibility for most foods and nutrients is considered good [27,28]. The FFQ includes 136 items divided into 9 food categories: dairy products, eggs, meat, fish, vegetables, fruits, legumes and cereals, oils and fats, pastries, beverages, and miscellaneous items. Participants indicated how often they consumed each food over the past year, specifying portion sizes with options ranging from “never or almost never” to “more than six times a day.” Spanish food composition tables were utilized to determine dietary consumption, considering the daily intake of each food and its nutritional composition [29,30]. An ad hoc computer system was employed to calculate the daily intake of each food by multiplying the typical serving size by the frequency of consumption.

### 2.3. Planetary Health Diet Assessment

For this analysis, we selected the Planetary Health Diet Index developed by Stubbendorf et al. [13] which has been adapted for the Spanish population (Table A1). We derived this Planetary Health Diet Index through the baseline FFQ and the 10-year FFQ.

This index defines target intakes and reference ranges and includes 14 food groups that are divided into “emphasized intake” and “limited intake”, drawing from prior descriptions provided by the Eat-Lancet Commission [12]. The emphasized foods are vegetables, fruits, unsaturated oils, legumes, nuts, whole grains, and fish, while the limited foods include beef and lamb, pork, poultry, eggs, dairy, potatoes, and added sugars. The score on the index for each item varies from 0 to 3 points depending on the amount consumed (g/d). For limited-intake foods, an inverse scoring system is used: 0 points indicate low adherence to the target for that item, while 3 points reflect high adherence. The index has a total possible score that spans from 0 to 42 points.

### 2.4. Assessment of Other Dietary Variables

We used two predefined indices to assess adherence to the Mediterranean pattern, which is primarily characterized by a high intake of plant-based foods such as olive oil, vegetables, legumes, fruits, nuts, whole grains, and fish as animal-based sources, as well as a moderate consumption of red wine, a low intake of lean meat and dairy products, and very low or no consumption of red and processed meat. We utilized the well-known Mediterranean Diet Score (MDS) developed by Trichopoulou et al. [31] with a total score ranging from 0 to 9 points. Additionally, we applied the 14-point Mediterranean Diet Adherence Screening (MEDAS) [32] used in the Prevención con Dieta Mediterránea (PREDIMED) trial, which includes items that are critical to an accurate assessment of the adherence to the traditional Mediterranean diet (MedDiet). Higher scores indicate greater adherence to the MedDiet.

### 2.5. Ascertainment of CVD

This study focused on CVD as the primary outcome and used self-reported questionnaires every two years to collect data. CVD events included myocardial infarction, CVD death, and stroke; all cases were validated through medical records. When a participant reported a CVD event, medical documentation was requested, and a team of cardiologists, unaware of the participants’ diets, evaluated the event. A non-fatal stroke was defined as a sudden neurological deficit lasting more than 24 h. Myocardial infarction diagnoses were based on universal criteria [33], and deaths were confirmed through certificates, medical records, and records linked to the National Institute of Statistics. Additionally, the National Death Index was consulted annually to identify participants who may have died in the cohort.

### 2.6. Co-Variables Evaluation

The additional covariates including sociodemographic information, anthropometric measurements, health habits, dietary intake, and lifestyle were gathered in the baseline Q0. The validity of the anthropometric information provided by the participants themselves (weight and height) has previously been analyzed in a subsample of the SUN cohort. BMI was estimated by dividing weight by height squared (kg/m^2^) and distributed into tertiles. Participants with chronic disease were recognized if they had a prior diagnosis or had received treatment with medications for managing the disease.

### 2.7. Statistical Analyses

Participants were categorized into the following quartiles (Q) according to their adherence to the Planetary Health Diet Index: Q1 or lowest adherence from 7 to 18 points; Q2 and Q3 or medium adherence from 19 to 21 and 22 to 23, respectively; and Q4 or highest overall adherence from 24 to 37 points. We calculated proportions for categorical variables and means along with standard deviations (SDs) for quantitative variables. To assess the relationship between the Planetary Health Diet Index quartiles and the incidence of CVD, we employed Cox proportional hazard regression models. Hazard ratios (HRs) were determined along with their 95% confidence intervals (CIs) for each quartile, using the first quartile (Q1) as the reference category. An HR greater than 1 indicates a higher risk of CVD, while an HR less than 1 suggests a lower likelihood of developing CVD. Based on existing evidence and previous findings of the SUN cohort, three multivariable-adjusted models were fitted, and age was used as a time variable in all Cox models. We adjusted our models as follows: Model 1 was adjusted for sex and stratified by age (deciles) and by year entering the cohort; Model 2 was additionally adjusted for total energy intake (kcal/d, continuous), smoking (never, current, and former smoker), educational level (years of higher education, continuous), alcohol intake (g/d, continuous), accumulated smoking habit (pack-years, continuous), physical activity (metabolic equivalent-h/week, continuous), body mass index (BMI [kg/m^2^, linear and quadratic terms, continuous]), snacking between meals (yes/no), watching television (h/d, continuous), time spent sitting (hours/week, continuous), and following a special diet at baseline (yes/no); lastly, Model 3 was additionally adjusted for hypertension (yes/no), family history of CVD (yes/no), and any diagnosis of diabetes (yes/no), hypercholesterolemia (yes/no), depression (yes/no), dyslipidemia (yes/no), and cancer (yes/no). Linear trend tests were used through quartiles, assigning the median value of each quartile, and treating the resulting variables as continuous. To reduce the impact of dietary variation, we applied time-dependent Cox models with repeated measurements using cumulative average dietary information from the components of the Planetary Health Diet. This updating of the dietary information was possible with a full repetition if the dietary assessment with an identical FFQ was performed after 10-year follow-up. In analyzing the repeated measures by the cumulative average method in a time-dependent Cox model, we calculated the mean between the baseline FFQ and the 10-year FFQ (i.e., cumulative average exposure) to provide a more accurate representation of the diet based on this dietary index.

To further assess the relationship between dietary patterns, the two-by-two correlation coefficients between the Planetary Health Diet and the Mediterranean Diet (using the MEDAS and MDS indices) were calculated. The following analyses and subgroup analyses were additionally performed to assess the robustness of our findings: (a) selection by sex, only men or women participants, (b) only participants <45 years or ≥45 years, (c) censoring participants at >50 years, (d) exclusion of participants with hypercholesterolemia and prevalent hypertension, (e) only health professionals or only non-health professionals participants, (f) exclusion of participants with prevalent cancer, (g) exclusion of participants who followed a special diet at baseline, (h) using different predefined energy intake limits (5th percentile and 95th percentile), (i) exclusion of participants with early CVD (≤2 years), and (j) exclusion of participants with ≥30 items missing in the FFQ. Statistical analyses were performed using STATA software (STATA version 14.1, StataCorp, College Station, TX, USA). All *p* values presented are two-tailed, and statistical significance was set at the conventional cut-off of *p* < 0.05.

## 3. Results

### 3.1. Characteristics of the Participants

A total of 18,656 participants were followed for a mean time of 11.5 years (minimum and maximum follow-up time is 0.04 and 19.64 years, respectively). During this period, 220 cases of prevalent CVD were identified, which included 95 cases of nonfatal acute myocardial infarction, 78 cases of nonfatal strokes, and 47 CVD deaths. Table 1 shows the total values of baseline socio-demographic characteristics of the participants and is categorized by quartiles of the Planetary Health Diet Index. Participants in the SUN cohort scored between 7 and 37 points on this index (score 0–42 points), with a mean score of 20.6 points (SD 3.4). The average age of the participants was 38 years (SD 12.1), and the mean baseline BMI was 23.5 kg/m^2^ (SD 3.5). Approximately 61% of the participants were women. Participants in the highest quartile of adherence to this diet (Q4) were more likely to have been former smokers, had accumulated more pack-years of smoking, were more physically active, followed a special diet, consumed supplements, had healthier dietary habits (higher MedDiet adherence and provegetarian scores), and had a higher prevalence of diseases. On the contrary, participants in Q1 were more likely to be single, to have never smoked, had more weight gain, and were more likely to snack between meals.

Participants who adhered more closely (Q4) to the Planetary Health Diet Index exhibited a higher consumption of vegetables, fruits, unsaturated oils, olive oil, legumes, nuts, whole grains, and fish; had higher carbohydrate and fiber intake; and had a healthier fat profile. Meanwhile, participants in Q1 had a higher consumption of foods recommended for limited consumption, such as beef, lamb, pork, poultry, eggs, dairy, potatoes, added sugars, and fast food. As for energy and nutrients, Q1 participants had the highest energy and fat intake, cholesterol consumption, and saturated fat consumption (Table 2).

### 3.2. Association Between Planetary Health Diet and CVD

Table 3 presents the results of the multivariate Cox regression analysis examining the association between adherence to the Planetary Health Diet Index and CVD risk. In the SUN cohort, the number of CVD cases across the quartiles of adherence to the diet were as follows: Q1 had 53 cases, Q2 78 cases, Q3 44 cases, and Q4 45 cases. The person-years calculated, representing the time at risk for participants, varied from 62,598 in Q1 to 41,549 in Q4, and mortality rates per 1000 person-years increased from Q1 (0.84) to Q4 (1.08). The analysis utilized three statistical models (Model 1, Model 2, and Model 3) to assess the relationship between diet adherence quartiles and CVD outcomes. The key findings from Model 3, which adjusted for various factors such as family history of CVD and prevalent conditions like diabetes, hypertension, and hypercholesterolemia (well-established CVD risk factors), showed no statistically significant association between adherence to the Planetary Health Diet and CVD risk. In comparison to the reference category (Q1), all point estimates appear to be protective (Model 3 Q2 HR 0.98, 95% CI 0.68–1.40; Q3 HR 0.74, 95% CI 0.49–1.12; Q4 HR 0.98, 95% CI 0.68–1.40); however, the p-for-trend value was 0.127, indicating that there was no significant linear trend across the quartiles regarding CVD risk.

To update the dietary analysis, we used time-dependent Cox regression models with repeated dietary measurements based on cumulative average data after 10 years of follow-up. Although the fully adjusted model showed an inverse relationship between the Planetary Health Diet Index and CVD risk, it was not statistically significant. The hazard ratios were 1.18 (95% CI 0.82–1.70) for updated dietary information and 0.82 (95% CI 0.55–1.23) when comparing participants in the highest versus lowest quartile in fully adjusted models (Table 3).

### 3.3. Correlation Between Planetary Health Diet and Mediterranean Diet

Our research explored the correlation between the Planetary Health Diet and MedDiet as measured by the MEDAS and MDS, and the correlation was 0.47 and 0.52, respectively.

### 3.4. Sensitivity Analyses

Multiple analyses were performed to corroborate our findings (Table 4). Overall, the results did not substantially change in any scenario or sub-group, observing 18,656 participants between the Planetary Health Diet Index and the incidence of CVD.

## 4. Discussion

The findings from the SUN cohort offer valuable insights into the relationship between adherence to the Planetary Health Diet and the risk of CVD among a cohort of Spanish university graduates. Despite the theoretical advantages of this dietary approach, particularly its capacity to mitigate CVD risk as noted in previous research, our research indicates no association.

However, the non-significant correlation found in this study does not undermine the well-known role of PBD in CVD, and in particular the role of the MedDiet [34]. Numerous studies have demonstrated a significant association between PBD and a reduced risk of CVD [8,9,35,36]. This dietary pattern, which emphasizes the consumption of fruits, vegetables, whole grains, legumes, nuts, and seeds while minimizing or excluding animal products, is rich in essential nutrients, fiber, and antioxidants, which collectively contribute to optimal cardiovascular health. Several reports have suggested that PBD can improve lipid profiles, lower blood pressure, reduce inflammation, and enhance glycemic control, all of which are crucial factors in preventing CVD [37]. The Planetary Health Diet, proposed by the EAT-Lancet Commission, is a sustainable eating plan designed to promote human health while minimizing environmental impact by focusing on plant-based foods [12]. The diet aims to balance nutritional needs with the preservation of natural resources. It also enhances nutrition and cardiovascular health by reducing processed foods, red meat, and unhealthy fats [22]. The improved health outcomes of the Planetary Health Diet are partly attributed to the nutrient-dense and bioactive compounds such as flavonoids and polyphenols, which also provide heart-protective benefits [38]. These bioactive compounds from fruits, vegetables, nuts, and whole grains have antioxidants, anti-inflammatory phytochemicals, and fiber, which help regulate glucose and LDL-cholesterol levels [39,40]. The diet’s low SFA content helps reduce the risk of atherosclerosis [41], while its high potassium levels aid in better blood pressure management [12]. In this study, participants with a higher adherence to the Planetary Health Diet Index consumed a healthier diet, including greater amounts of vegetables, fruits, unsaturated oils, olive oil, legumes, nuts, cereals, whole grains, fish, and fiber; nonetheless, the findings were not significant, and these results were consistent in analyses under different scenarios. It has to be taken into consideration that the Planetary Health Diet Index does not sufficiently assess adherence to the MedDiet, which is well known for its health benefits. In fact, in our study, the correlation between both dietary patterns was moderate. Therefore, individuals may still benefit from the MedDiet. Another potential reason for not observing an association could be the minimal differences in dietary intake between the quartiles being compared. The consumption of MUFA was quite similar in both the lowest quartile (Q1) and the highest quartile (Q4), as was the intake of PUFA. Additionally, the percentage of total fat intake showed little variation across quartiles. Since these food components are known to be related to CVD, this lack of significant variation in key dietary components may have contributed to the absence of a clear association in the study findings.

Previous investigations examining the link between the Planetary Health Diet and CVD have found either inverse or no significant associations. In line with our study, the NutriNet-Santé cohort, a Mediterranean cohort with 786 cases of CVD events with 12 years of follow-up found no association between the EAT-Lancet diet (range −162 to 332 points) and CVD [1]. Lazarova et al. [11] did not find an association between adherence to the Planetary Health Diet and all-cause stroke. Also, in a study conducted with the UK Biobank cohort, which used an index ranging from 0 to 11 points to assess adherence to the EAT-Lancet diet, no association was found between the diet and CVD [23]. On the other hand, certain studies have found a beneficial association between the Planetary Health Diet or EAT-Lancet diet and CVD [2,15,16,17,18,20,21,24]. The EPIC-Oxford cohort and the EPIC-Netherlands cohort found a lower risk of ischemic heart disease, CVD, and coronary heart disease with higher adherence to the diet compared with the lowest adherence [18,21]. Particularly, in British adults from the EPIC-Oxford cohort, it is noteworthy that better adherence to the EAT-Lancet diet (range 0–14 points) was associated with a 28% risk reduction in ischemic heart disease [21], while in the Dutch population from the EPIC-Netherlands cohort, the risk reduction was 14% [18], in accordance with the findings from another study conducted in the UK Biobank cohort [16]. The findings of two additional studies conducted within the same UK Biobank cohort, which employed an index with a broader scoring range (0 to 130 points) that assessed 14 dietary components classified into three categories, adequacy, optimal, and moderation intake, revealed a positive correlation between adherence to the diet and CVD [16,22]. In the study conducted by Zhang et al. [15], the EAT-Lancet diet index (range 0–42 points) was evaluated using only a single dietary assessment at baseline, without being updated throughout the extended follow-up period. Such different results could be partially attributed to the absence of a standardized, universal operational definition of the EAT-Lancet diet and the variety of scores and methodologies to measure adherence to this novel dietary pattern [42].

It must be highlighted that our study distinguishes itself from previous research in several significant ways. First, studies typically employed at least two 24 h dietary assessments alongside a touchscreen FFQ. In the SUN cohort, dietary information was collected through a validated self-reported FFQ. Second, the scoring method for the Planetary Health Diet in previous studies differed considerably from our approach. For instance, Sotos-Prieto et al. [16] and Colizzi et al. [18] used a continuous scoring system that enabled them to capture variations relative to reference dietary levels, whereas our score is ordinal, which may limit our ability to detect associations effectively. Furthermore, variations in cohort profile, sample size, the number of CVD cases, and baseline characteristics of participants, diet assessment, dietary scores construction, follow-up duration, food group definitions or food questionnaires likely contribute to the discrepancies in the study results. Concerning the populations and study designs, the difference in scoring criteria may also contribute to the discrepancy in results. It is recognized that dietary scoring systems for the calculations showed varying odds of CVD and its sub-types [43]. The Stubbendorff index [13] has several advantages including the use of an ordinal scale system to score each dietary component, which makes it simple to use and interpret. Thus, it is necessary for more epidemiological and experimental research to refine the methodology for calculating the scores concerning CVD and mortality [44].

Without a standardized and universally accepted score, it is unavoidable that researchers interpret and implement the Planetary Health Diet in varying ways.

### Strengths and Limitations

Several limitations of this study should be acknowledged. First, our participants were highly educated. Only individuals with a university degree were included. Thus, they have an overall higher health literacy and healthier lifestyle (higher physical activity and better working conditions), which may result in findings that are not fully representative of the general population. In particular, the EPIC- Netherlands combines two cohorts with more than 40,000 participants, which are comparatively younger and with a low proportion of participants with a high educational level, and found a lower risk or ischemic heart disease, CVC, and coronary heart disease. Nevertheless, the applicability of the results should be grounded in biological mechanisms rather than statistical representation. Second, the FFQ was self-reported, which may lead to measurement errors. However, the FFQ remains a widely recognized tool in nutritional epidemiology for evaluating dietary habits and has undergone multiple validations [28]. Third, despite adjusting our models for conventional CVD risk factors, residual confounding could not be entirely ruled out. Fourth, we were unable to independently assess the relationship between the Planetary Health Diet Index and fatal or non-fatal myocardial infarction, stroke, and deaths from CVD due to the small number of cases in our cohort, which limited component-specific analyses for the composite CVD outcome. Lastly, the Planetary Health Diet Index has yet to undergo formal validation. However, this index is based on the best scientific evidence, and it was previously used in other studies [13].

The strengths of this study include its large sample size, extended follow-up period, high retention rate (91%), and evaluation of Planetary Health Diet at baseline and after 10 years of follow-up by the cumulative average method, trying to capture the relationship of a time-varying exposure variable, such as diet, on the development of CVD, and the results barely changed. Cardiovascular events were confirmed blindly through medical records, reducing the risk of misclassification bias. Additionally, this study was able to account for a wide range of potential confounders. Extensive analyses under different assumptions and confirmation of deaths through the Spanish National Death Index further strengthen this study’s reliability.

## 5. Conclusions

In a large cohort of Spanish adults, greater adherence to a Planetary Dietary Pattern, as recommended by the EAT-Lancet Commission to promote both human health and environmental sustainability, documented no significant associations with a reduced risk of CVD. This diet, while promising in its approach, does not adequately measure adherence to the MedDiet, a well-established dietary pattern known for its health benefits, particularly in reducing CVD risk. This suggests that individuals may still benefit from adhering to the MedDiet, which emphasizes a balanced intake of plant-based foods, and healthy fats. Further research may be necessary to clarify the relationship between the newer dietary framework, the Planetary Health Diet, and CVD risk.

## Figures and Tables

**Figure 1 nutrients-17-00027-f001:**
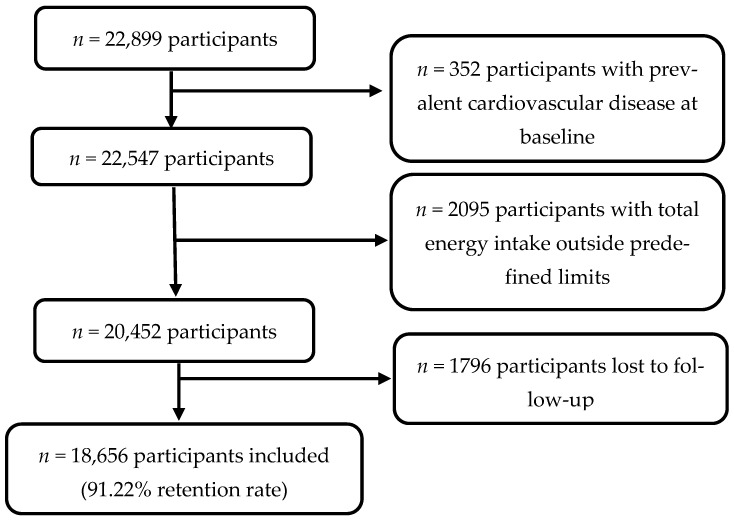
Flowchart depicting the selection of participants in the SUN project (1999–2020) included in the present analyses.

**Table 1 nutrients-17-00027-t001:** Baseline characteristics of participants according to quartiles of the adherence to the Planetary Health Diet in the SUN project ^1^.

	Q1	Q2	Q3	Q4	Total Sample
*n* (frequency)	4758	6895	3706	3297	18,656
Planetary Health Diet range	7–18	19–21	22–23	24–37	
Planetary Health Diet median	17	20	22	24	20.6
Age	38.2 (12.4)	38 (12.1)	38.1 (12.2)	38 (12.3)	38 (12.1)
Marital status (%)					
Single	50.4	45.1	40.2	38.7	44.4
Married	45.2	49.7	54.3	53.3	50.1
Other	4.6	5.2	5.5	1.0	5.5
Smoking (%)					
Never	51.9	50.1	46.3	46.2	49.1
Current smoker	23.7	22.4	21.7	18.3	21.9
Former smoker	24.4	27.5	32.0	35.5	29.0
Cumulative smoking habit (pack-years)	5.4 (9.3)	5.7 (9.4)	6.6 (10.2)	7.2 (10.8)	6.1 (9.8)
Years of university	5.1 (1.5)	5.0 (1.5)	5.1 (1.5)	5.1 (1.5)	5.1 (1.5)
Physical activity (METs/h/week)	20.1 (22.2)	21.5 (22.3)	22.5 (23.8)	24.2 (24.2)	21.8 (22.9)
BMI (kg/m^2^)	23.6 (3.4)	23.5 (3.5)	23.7 (3.7)	23.5 (3.5)	23.5 (3.5)
Time spent sitting (h/d)	5.5 (2.0)	5.3 (2.1)	5.2 (2.0)	5.1 (2.1)	5.3 (2.1)
Watching television (h/d)	1.6 (1.2)	1.6 (1.2)	1.6 (1.2)	1.5 (1.2)	1.6 (1.2)
Weight gain of >3 kg in the last 5 years (%)	34.5	30.1	29.7	24.4	30.1
Snacking between meals (%)	37.4	33.7	30.5	29.4	33.2
Follow-up of special diet (%)	5.7	7.2	8.7	13.4	8.2
Supplements consumption (%)	17.5	18.1	18.6	22.4	18.8
Prevalent diseases (%)					
Diabetes	1.0	1.5	1.9	3.3	1.8
Hypertension	8.5	9.8	12.5	12.6	10.5
Dyslipidemia	5.8	6.0	7.4	8.1	6.6
Depression	10.6	11.1	11.5	13.8	11.5
Cancer	2.2	2.4	2.8	3.7	2.6
Trichopoulou MDS [31], (range, 0–9)	3.0 (1.5)	4.1 (1.6)	4.8 (1.6)	5.6 (1.5)	4.2 (1.8)
MEDAS [32], (range, 0–14)	4.9 (1.6)	5.9 (1.6)	6.4 (1.7)	7.3 (1.8)	6 (1.8)
Provegetarian score, (range, 12–60)	32.2 (4.3)	35.4 (4.0)	37.8 (3.9)	40.8 (4.2)	36 (5)

^1^ Values are expressed as means (SD) or percentages. *METs*: metabolic equivalents, *BMI*: body mass index, *MDS*: Mediterranean Diet Score, and *MEDAS*: Mediterranean Diet Adherence Screener.

**Table 2 nutrients-17-00027-t002:** Dietary baseline characteristics according to quartiles of the Planetary Health Diet among participants in the SUN cohort.

	Q1	Q2	Q3	Q4
*n* (frequency)	4758	6895	3706	3297
Planetary Health Diet range	7–18	19–21	22–23	24–37
Food (g/d)				
Vegetables	394.7 (268.9)	528.9 (314.3)	583.9 (335.8)	667.1 (415.3)
Fruits	171.9 (164.6)	276.5 (219.2)	347.4 (267.6)	412.2 (316.6)
Unsaturated oils ^1^	17.6 (13.6)	22.3 (15.6)	25.2 (16.7)	28.2 (18.3)
Olive oil	13.3 (11.7)	17.9 (14.0)	21.1 (15.5)	24.4 (17.1)
Legumes	18.8 (13.7)	22.1 (15.8)	24.1 (17.7)	28.8 (25.4)
Nuts	4.6 (6.1)	6.0 (8.6)	7.8 (11.7)	14.6 (19.8)
Cereals	97.0 (70.7)	100.2 (71.0)	101.8 (70.4)	110.8 (78.6)
Whole grains	5.4 (15.0)	9.9 (24.1)	15.3 (33.2)	29.0 (49.0)
Fish	84.3 (55.1)	97.9 (55.5)	104.8 (58.7)	111.4 (73.6)
Beef and lamb	62.8 (33.8)	56.6 (33.0)	52.0 (34.3)	37.9 (34.9)
Pork	81.9 (43.8)	75.5 (41.9)	67.0 (41.0)	52.1 (41.1)
Poultry	51.9 (37.6)	43.8 (33.0)	36.2 (29.6)	29.8 (29.9)
Eggs	28.0 (17.6)	24.7 (14.6)	21.0 (15.2)	15.6 (12.3)
Dairy	506.7 (284)	431.0 (249)	381.0 (242.6)	328.5 (229.7)
Potatoes	69.7 (51.8)	54.3 (43.5)	44.2 (38.0)	37.0 (33.1)
Added sugars	58.3 (30.3)	45.7 (24.9)	37.8 (22.4)	30.9 (21.0)
Fast food ^2^	28.1 (23.3)	22.7 (19.8)	18.7 (19.2)	15.3 (17.6)
Energy and nutrients				
Energy (kcal/d)	2434 (651)	2352 (610)	2277 (582)	2248 (586)
Carbohydrates (% TEI)	42.7 (7.1)	43.1 (7.0)	43.5 (7.5)	45.1 (8.6)
Fiber (g/d)	17.7 (7.3)	21.8 (8.3)	24.3 (9.5)	29.2 (12.1)
Proteins (% TEI)	18.5 (3.5)	18.5 (3.3)	18.3 (3.1)	17.5 (3.4)
Fats (% TEI)	36.9 (6.0)	36.4 (6.3)	36.1 (6.8)	35.2 (7.7)
SFAs	13.8 (3.1)	12.8 (3.0)	12.0 (2.9)	10.6 (3.1)
TFAs	0.4 (0.2)	0.4 (0.2)	0.3 (0.2)	0.3 (0.2)
PUFAs	5.3 (1.6)	5.1 (1.5)	5.1 (1.6)	5.2 (1.7)
n-3 fatty acids (g/d)	2.5 (1.3)	2.6 (1.2)	2.5 (1.1)	2.6 (1.2)
n-6 fatty acids (g/d)	20.0 (13.6)	18.0 (11.8)	16.7 (10.89	15.6 (11.3)
MUFAs	15.4 (3.1)	15.7 (3.6)	16.0 (4.0)	16.2 (4.5)
Cholesterol (mg/d)	473.4 (155.7)	432.4 (142.4)	388.1 (128.7)	325.4 (126.9)
Alcohol (g/d)	6.3 (10.2)	6.7 (10.3)	6.9 (10.0)	6.6 (9.6)

^1^ Including olive oil. ^2^ Including hamburgers, pizza, and sausages. % *TEI*, total energy intake; *SFA*, saturated fatty acids; *TFA*, trans fatty acids; *PUFA*, polyunsaturated fatty acids; *n*-*6 fatty acids*, omega 6 fatty acids; *n-3 fatty acids*, omega-3 fatty acids; *MUFA*, monounsaturated fatty acids.

**Table 3 nutrients-17-00027-t003:** Hazard ratios (HRs) and 95% confidence intervals (CIs) for the association between Planetary Health Diet and the risk of cardiovascular disease at baseline and after 10 years of follow-up in 18,656 participants in the SUN cohort.

	Q1	Q2	Q3	Q4	*p* for Trend
*n*	4758	6895	3706	3297	
PHD range	7–18	19–21	22–23	24–37	
CVD	53	78	44	45	
Person-years	62,597.7	89,631.6	48,404.5	41,548.6	
Mortality rate/1000 person years	0.84	0.87	0.91	1.08	
Model 1	1 (Ref.)	0.98 (0.69–1.40)	0.75 (0.50–1.13)	0.74 (0.49–1.12)	0.081
Model 2	1 (Ref.)	0.97 (0.68–1.39)	0.76 (0.50–1.15)	0.78 (0.52–1.19)	0.150
Model 3	1 (Ref.)	0.98 (0.68–1.40)	0.74 (0.49–1.12)	0.77 (0.51–1.18)	0.127
Cumulative Diet Average ^1^					
Planetary Health Diet range	7–19	19.5–21	21.5–23	23.5–36.5	
CVD cases	61	63	49	47	
Person-years	77,681.8	66,065.4	542,92.8	54,140.7	
Mortality rate/1000 person years	0.79	0.95	0.90	1.04	
Model 3	1 (Ref.)	1.18 (0.82–1.70)	0.88 (0.60–1.31)	0.82 (0.55–1.23)	0.229

^1^ Repeated measures: cumulative average information of the Planetary Health Diet Index at baseline and after 10 years of follow-up. Ref. reference. Model 1 was adjusted for sex and stratified by age (deciles) and by year entering the cohort. Model 2 was additionally adjusted for total energy intake (kcal/d, continuous), smoking (never, current, and former smoker), educational level (years of higher education, continuous), alcohol intake (g/d, continuous), accumulated smoking habit (pack-years, continuous), physical activity (metabolic equivalent-h/week, continuous), body mass index (BMI [kg/m^2^, linear and quadratic terms, continuous]), snacking between meals (yes/no), watching television (h/d, continuous)**,** time spent sitting (hours/week, continuous), and following a special diet at baseline (yes/no). Model 3 was additionally adjusted for hypertension (yes/no), family history of CVD (yes/no), and any diagnosis of diabetes (yes/no), hypercholesterolemia (yes/no), depression (yes/no), dyslipidemia (yes/no), and cancer (yes/no).

**Table 4 nutrients-17-00027-t004:** Hazard ratios (HRs) and 95% confidence intervals (CIs) for the association between Planetary Health Diet and the risk of cardiovascular disease in the SUN cohort. Quartile 4 vs. Quartile 1.

	N	CVD Events	HR (95%CI)	*p* for Trend
Main analyses				
Main analyses	18,656	220	0.77 (0.51–1.18)	0.127
Sensitivity analyses				
Only men	11,312	45	0.68 (0.24–1.92)	0.450
Only women	7344	175	0.75 (0.47–1.21)	0.151
Only <45 years	13,234	41	0.86 (0.30–2.47)	0.680
Only ≥45 years	5422	179	0.80 (0.50–1.27)	0.170
Excluding hypertension or hypercholesterolemia at baseline	14,278	86	0.81 (0.41–1.57)	0.420
Only health professionals	11,980	134	0.75 (0.44–1.28)	0.676
Only non-health professionals	6676	86	0.71 (0.36–1.44)	0.312
Excluding participants with cancer at baseline	18,163	211	0.77 (0.50–1.17)	0.110
Excluding participants with special diet at baseline	17,129	192	0.75 (0.48–1.18)	0.910
Energy limits: percentiles 5–95 at baseline	18,517	206	0.70 (0.45–1.09)	0.071
Excluding early cases (first 2 years)	18,628	192	0.76 (0.49–1.19)	0.910
Excluding participants with 30 or more missing values in FFQ	17,453	182	0.79 (0.50–1.25)	0.133

Adjusted for sex and stratified by age (deciles), by year entering the cohort total energy intake (kcal/d, continuous), smoking (never, current, and former smoker), educational level (years of higher education, continuous), alcohol intake (g/d, continuous), accumulated smoking habit (pack-years, continuous), physical activity (metabolic equivalent-h/week, continuous), body mass index (BMI [kg/m^2^, linear and quadratic terms, continuous]), snacking between meals (yes/no), watching television (h/d, continuous)**,** time spent sitting (hours/week, continuous), following a special diet at baseline (yes/no), hypertension (yes/no), family history of CVD (yes/no), and any diagnosis of diabetes (yes/no), hypercholesterolemia (yes/no), depression (yes/no), dyslipidemia (yes/no), and cancer (yes/no).

## Data Availability

Information and data from this study are available on request from SUN Scientific Committee sun@unav.es.

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
