# Peer review of "Planetary Health Diet and Cardiovascular Disease Risk in the Seguimiento Universidad de Navarra (SUN) Cohort"

_nutrients, 2024, doi:10.3390/nu17010027_

Round 1
Reviewer 1 Report
Comments and Suggestions for Authors
Sustainability is a topic that interests many readers and this paper will be of interest for a majority of people.
Corrections/suggestions follow:
- It seems that you forgot to add the last author as in line 5 there is an "and" with affiliation #5
- It seems that you have self-reference of 17%. Try reducing it.
Reviewer 2 Report
Comments and Suggestions for Authors
This article is valuable and scientifically sound, with thorough statistical analysis conducted using Cox regression, clearly presented in well-organized tables. It is characterized by a large sample size and extended follow-up period. The study uniquely combines aspects of nutrition and sustainability, setting it apart in the field.
Although the questionnaires used were validated, relying on self-reported dietary data introduces the risk of subjective bias. Certain phrases, such as "statistically significant" and "sensitivity analyses" are repeated frequently throughout the text. Revising the text to reduce redundancy would enhance readability. "Evidence has shown" can be rephrased, for example, as "Research indicates" to improve stylistic variety.
The references are relevant, and the text shows no signs of plagiarism. I recommend the article for publication, although minor improvements in the English language could make it more accessible and easier to read.
Author Response
|
Comments 1: This article is valuable and scientifically sound, with thorough statistical analysis conducted using Cox regression, clearly presented in well-organized tables. It is characterized by a large sample size and extended follow-up period. The study uniquely combines aspects of nutrition and sustainability, setting it apart in the field. The references are relevant, and the text shows no signs of plagiarism.
|
|
Response 1: We thank you for reviewing our manuscript and for your suggestions, which helped us to improve it.
|
|
Comments 2: Although the questionnaires used were validated, relying on self-reported dietary data introduces the risk of subjective bias. Certain phrases, such as "statistically significant" and "sensitivity analyses" are repeated frequently throughout the text. Revising the text to reduce redundancy would enhance readability. "Evidence has shown" can be rephrased, for example, as "Research indicates" to improve stylistic variety. |
|
Response 2: Thank you very much for your suggestion, we have rephrased several sentences in the new version of manuscript:
Lines 21-23: After a median follow-up time of 11.5 years, 220 cases of CVD were identified. Higher adherence to the Planetary Health Diet revealed no significant reduction in CVD risk associated with the diet
Lines 281-283: Despite the theoretical advantages of this dietary approach, particularly its capacity to mitigate CVD risk as noted in previous research, our research indicated no association.
Lines 290-291: Several reports have suggested that PBD can improve lipid profiles, lower blood pressure, reduce inflammation
Lines 306-307: nonetheless, the findings were not significant and these results were consistent in analyses under different scenarios.
Lines 390-392: Extensive analyses under different assumptions and confirmation of deaths through the Spanish National Death Index further strengthen the study's reliability.
Comments 3: Although the questionnaires used were validated, relying on self-reported dietary data introduces the risk of subjective bias. Certain phrases, such as "statistically significant" and "sensitivity analyses" are repeated frequently throughout the text. Revising the text to reduce redundancy would enhance readability. "Evidence has shown" can be rephrased, for example, as "Research indicates" to improve stylistic variety.
Response 3: Thank you very much for your suggestion, we have rephrased several sentences in the new version of manuscript:
Lines 21-23: After a median follow-up time of 11.5 years, 220 cases of CVD were identified. Higher adherence to the Planetary Health Diet revealed no significant reduction in CVD risk associated with the diet
Lines 281-283: Despite the theoretical advantages of this dietary approach, particularly its capacity to mitigate CVD risk as noted in previous research, our research indicated no association.
Lines 290-291: Several reports have suggested that PBD can improve lipid profiles, lower blood pressure, reduce inflammation
Lines 306-307: nonetheless, the findings were not significant and these results were consistent in analyses under different scenarios.
Lines 390-392: Extensive analyses under different assumptions and confirmation of deaths through the Spanish National Death Index further strengthen the study's reliability.
|
|
4. Response to Comments on the Quality of English Language |
|
Point 1: I recommend the article for publication, although minor improvements in the English language could make it more accessible and easier to read. |
|
Response 4: Thank you very much for your proposal. The new version of the manuscript has been revised by all authors to comply with your suggestion. Corrections of typos and repetitions, and grammatical improvements have been made.
|
Reviewer 3 Report
Comments and Suggestions for Authors
The manuscript focuses on the association between planetary health diet and cardiovascular disease. To investigate this, a large cohort consisting of university graduates from Spain was used. The manuscript is well-written and follows a clear structure. Nonetheless, I have some comments that may help to strenghten the manuscript.
- It was unclear to me, how data from a ten-year-follow-up can be used when the baseline questionnaire Q0 was completed till 2022 (line 79). Please clarfiy this in the manuscript.
- Please clarify if planetary diet was only assessed during follow-up or it it was included in the baseline questionnaire.
- Please include min and max for the follow-up time (line 204).
- Table 1: Please include also a column for the total values in the data set.
- The authors may discuss whether they could not find a significant assoication because the sample is very homogenous. Only individuals with a university degree were included who have an overall higher health literacy and healthier lifestyle (higher physical activity, better working conditions, and so on). This aspect should not only be discussed in the limitation section but may also explain why other studies found an association in their more heterogenous samples.
Author Response
|
Comments 1: The manuscript focuses on the association between planetary health diet and cardiovascular disease. To investigate this, a large cohort consisting of university graduates from Spain was used. The manuscript is well-written and follows a clear structure.
|
|
Response 1: We thank you for reviewing our manuscript and for the suggestions, which helped us to improve it. |
|
Comments 2: Nonetheless, I have some comments that may help to strength in the manuscript. It was unclear to me, how data from a ten-year-follow-up can be used when the baseline questionnaire Q0 was completed till 2022 (line 79). Please clarify this in the manuscript.
|
|
Response 2: We have further clarified our methodology for leveraging repeated measurements of dietary intake. Our study has a main aim analysis and a secondary analysis.
The main aim was to investigate the baseline association between adherence to the Planetary Health Diet and the risk of CVD in the SUN project. We have rephrased the following paragraph in the Material and Methods section as follows, and we have corrected the information in Figure 1
Lines 79-85: For this study, a total of 22,899 participants recruited before September 2020 were eligible for the analysis of incident CVD, ensuring that they were able to complete at least the first follow-up questionnaire. We excluded 352 participants with pre-existing CVD, as well as 2,095 participants who fell outside predefined limits of total energy intake (<800 kcal/d or >4,200 kcal/d for men and <500 kcal/d or >3,500 kcal/d for women) [26] and 1,796 participants without follow-up. As a result, 18,656 participants were included in our final analysis (Figure 1).
On the other hand, in a secondary analysis, aiming to reduce the impact of dietary variation, we applied time-dependent Cox models with repeated measurements using cumulative average dietary information from the components of the Planetary Health Diet. This updating of the dietary information was possible given with a full repetition if the dietary assessment with an identical FFQ was performed after 10-year follow-up. In analyzing the repeated measures by cumulative average method in a time-dependent Cox model, we calculated the mean between the baseline FFQ and the 10-year FFQ (i.e., cumulative average exposure) to provide a more accurate representation of the diet based on this dietary index. In cases where a participant's 10-year FFQ data was missing, the corresponding value was substituted with the value from their baseline FFQ. This approach was implemented to ensure continuity and consistency in the data analysis.
Lines 189-197: To reduce the impact of dietary variation, we applied time-dependent Cox models with repeated measurements using cumulative average dietary information from the components of the Planetary Health Diet. This updating of the dietary information was possible given with a full repetition if the dietary assessment with an identical FFQ was performed after 10-year follow-up. In analyzing the repeated measures by the cumulative average method in a time-dependent Cox model, we calculated the mean between the baseline FFQ and the 10-year FFQ (i.e., cumulative average exposure) to provide a more accurate representation of the diet based on this dietary index.
To clarify this issue in the new version of manuscript, we have added a new strength in the Strengths and limitations section as follows:
Lines 384-388: The strengths of this study include its large sample size, extended follow-up period, high retention rate (91%) and evaluation of Planetary Health Diet at baseline and after 10 y of follow-up by cumulative average method, trying to capture the relationship of a time-vary in exposure, such as diet, on the development of CVD, and the results barely changed.
Comments 3: Please clarify if planetary diet was only assessed during follow-up or it was included in the baseline questionnaire. Response 3: Please check our answer to your previous question. In addition, we also included the following sentences to further clarify this issue.
Lines 122-124: For this analysis, we selected the Planetary Health Diet Index developed by Stubbendorf et al. [13] which has been adapted for the Spanish population (Table S1). We derived this Planetary Health Diet Index through the baseline FFQ and the 10-year FFQ. |
|
|
|
|
|
Comments 4: Please include min and max for the follow-up time (line 204). Response 4: We have clarified this information in the new version of the manuscript.
Lines 214-215: A total of 18,656 participants were followed for a mean time of 11.5 years (minimum and maximum follow-up time is 0.04 and 19.64, respectively). |
|
|
Comments 5: Table 1: Please include also a column for the total values in the data set.
Response 5: OK. We have included it in Table 1 in the revised version of the manuscript.
Comments 6: The authors may discuss whether they could not find a significant association because the sample is very homogenous. Only individuals with a university degree were included who have an overall higher health literacy and healthier lifestyle (higher physical activity, better working conditions, and so on). This aspect should not only be discussed in the limitation section but may also explain why other studies found an association in their more heterogenous samples.
Response 6: We value your suggestion. We have rewritten some other parts in the manuscript to better explain this issue.
Lines 322-325: Lazarova et al. [11] did not find an association between adherence to the Planetary Health Diet and all-cause stroke. Also, in this particular study conducted with the UK Biobank cohort, which used an index ranging from 0 to 11 points to assess adherence to the EAT-Lancet diet, no association was found between the diet and CVD [23].
Lines 332-338: the Dutch population from the EPIC-Netherlands cohort the risk reduction was 14% [18] in accordance with the findings from other study conducted in the UK Biobank cohort [16]. The findings of two additional studies conducted within the same UK Biobank cohort, employed an index with a broader scoring range (0 to 130 points) that assessed 14 dietary components classified into three categories: adequacy, optimal, and moderation intake, revealed a positive correlation between adherence to the diet and CVD [16,22].
Lines 348-361: For instance, Sotos-Prieto et al. [16] and Colizzi et al. [18] used a continuous scoring system that enabled them to capture variations relative to reference dietary levels, whereas our score is ordinal, which may limit our ability to detect associations effectively. Furthermore, variations in cohort profile, sample size, variations in the number of CVD cases and baseline characteristics of participants, diet assessment, dietary scores construction, follow-up duration, food group definitions or food questionnaires, likely contribute to the discrepancies in the study results. Concerning to populations and study designs, the difference in scoring criteria may also contribute to the discrepancy in results. It is recognized that dietary scoring systems for calculation showed varying odds of CVD and its sub-types [43]. Stubbendorff index [13], has several advantages including the use of an ordinal scale system to score each dietary component, which makes it simple to use and interpret. Thus, it is necessary more epidemiological and experimental research to refine the methodology for calculating the scores concerning cardiovascular diseases and mortality [44].
Lines 366-372: Only individuals with a university degree were included. Thus, they have an overall higher health literacy and healthier lifestyle (higher physical activity and better working conditions), which may result in findings that are not fully representative of the general population. In particular, and the EPIC- Netherlands combines two cohorts with more than 40,000 participants, comparative younger and with low proportion participants with of high educational level, found a lower risk or ischaemic heart disease, CVC, and coronary heart disease.